# Long-Term Survival in BRCA1 Mutant Advanced Ovarian Cancer: Unveiling the Impact of Olaparib

**DOI:** 10.3390/diagnostics14171898

**Published:** 2024-08-29

**Authors:** Vlad-Adrian Afrăsânie, Alexandra Rusu, Adelina Silvana Gheorghe, Eliza Maria Froicu, Elena Adriana Dumitrescu, Bogdan Gafton, Teodora Alexa-Stratulat, Lucian Miron, Dana Lucia Stănculeanu, Mihai Vasile Marinca

**Affiliations:** 1Department of Medical Oncology, Regional Institute of Oncology, 700483 Iasi, Romania; adrian.afrasanie@umfiasi.ro (V.-A.A.); rusu.alexandra@email.umfiasi.ro (A.R.); eliza-maria.froicu@umfiasi.ro (E.M.F.); gaftonbogdan@yahoo.com (B.G.); teodora_alexa@yahoo.com (T.A.-S.); lucmir@gmail.com (L.M.); mihai.marinca@umfiasi.ro (M.V.M.); 2Department of Oncology, Faculty of Medicine, “Grigore T. Popa” University of Medicine and Pharmacy, 700115 Iasi, Romania; 3Department of Oncology, “Carol Davila” University of Medicine and Pharmacy, 020021 Bucharest, Romania; elena-adriana.dumitrescu@drd.umfcd.ro (E.A.D.); dana.stanculeanu@umfcd.ro (D.L.S.); 4Department of Medical Oncology I, Institute of Oncology “Prof. Dr. Al. Trestioreanu” Bucharest, 022328 Bucharest, Romania

**Keywords:** BRCA1 mutation, ovarian cancer, Olaparib, chemotherapy

## Abstract

Ovarian cancer is one of the most frequent malignancies in women. The treatment landscape underwent significant changes as new agents were introduced in ovarian cancer management over the last decade. We present two cases of long responses to Olaparib in BRCA (BReast CAncer gene) mutant ovarian cancer patients. The first case belongs to a 42-year-old female diagnosed with advanced ovarian carcinoma with a rare germinal mutation (BRCA1 c.68_69delAG, commonly found in descendants of Ashkenazi Jewish populations, but also Arabic and Asian ones) and a significant family history of ovarian and breast cancers. After poorly tolerated neoadjuvant chemotherapy, the patient underwent total hysterectomy, bilateral adnexectomy, and intraperitoneal hyperthermic chemotherapy. After eight months, the disease progressed, and first-line platinum chemotherapy was administered. Although not well-tolerated (grade 3 anemia, allergic reactions), chemotherapy resulted in a partial response, and given the patient’s characteristics, maintenance with Olaparib was recommended. Treatment is ongoing (total current duration 69 months) and tolerated well (grade 1 side effects). This case illustrates the long-term benefits that novel therapies like Olaparib may offer in patients with platinum-sensitive relapsed ovarian cancer harboring a rare BRCA mutation. The second case highlights a 55-year-old postmenopausal woman diagnosed with ovarian cancer, FIGO stage IVA. Initial treatment included six cycles of chemotherapy, which led to a partial response, followed by interval debulking surgery and another four cycles of chemotherapy. Subsequent Olaparib maintenance therapy post BRCA1 mutation identification contributed to a significant progression-free survival of 65 months until disease recurrence and secondary cytoreductive surgery, showcasing the effectiveness of PARP inhibitors in personalized oncology treatment of ovarian cancer.

## 1. Introduction

Ovarian cancer is the fifth most common neoplasia in women worldwide and the most lethal gynecologic malignancy. The poor prognosis and low five-year survival rate are attributed to non-specific symptoms leading to late diagnosis [1].

The treatment landscape underwent significant changes as new agents were introduced in ovarian cancer management over the last decade [2,3]. One particularly relevant direction was poly ADP-ribose polymerase (PARP) targeted blockage, which prevents the repair of chemotherapy-damaged DNA, resulting in enhanced cell death. Today, PARP inhibitors administered as maintenance therapy after cytotoxic agents have become a standard of care in BReast CAncer gene (BRCA) 1/2 deficient ovarian tumors. They are showing promise in other BRCA-associated malignancies as cancer treatment moves toward the proteomics and genomics era [4,5,6,7]. While the effectiveness of Olaparib in ovarian cancer cases is well-supported by the existing literature, the significance of these new case reports lies in the detailed presentation of two Romanian cases, which contribute valuable information to the growing body of data on the prolonged efficacy of PARP inhibitors in diverse patient populations. Moreover, the specific BRCA1 mutations observed in these patients, combined with their extended response to treatment, offer valuable insights that could contribute to personalized treatment strategies and further research into the genetic factors influencing PARP inhibitor efficacy.

## 2. Case Presentations

### 2.1. First Case Presentation

#### 2.1.1. Main Characteristics and Admission to Hospital

This report includes a case of a 42-year-old premenopausal female presenting with an enlarged abdomen and asthenia; an ultrasound scan showed tumors in both ovaries.

#### 2.1.2. History, Comorbidities, and Interventions

The obstetrical history of the patient included no births and no abortions. Her mother had been diagnosed with ovarian cancer at the age of 49, with fast progression leading to rapid death; her grandmother had breast cancer at the age of 52, but the evolution of the disease could not be specified.

No other relevant conditions were reported.

#### 2.1.3. Examinations and Investigations

In June 2015, the patient was admitted to the Iasi Regional Oncology Institute (IRO). A computed tomography (CT) scan of the thorax, abdomen, and pelvis showed tumors in both ovaries (right, 92/57/65 mm; left, 70/50/72 mm) and one 40/38 mm left iliac adenopathy (Figure 1).

#### 2.1.4. Diagnosis

Diagnostic laparoscopy revealed bilateral ovarian tumors, peritoneal carcinomatosis, and ascites (stage IIIB, cT3bN0M0). Biopsy from the ovarian masses and peritoneal nodules showed the infiltration of high-grade serous ovarian carcinoma.

#### 2.1.5. Treatment and Outcomes

The Multidisciplinary Tumor Board decision favored primary chemotherapy. In July 2015, the patient was admitted to the IRO (Regional Institute of Oncology in Iași, Romania) Medical Oncology Department, presenting with asthenia, pallor, and ascites. Her performance status was assessed as grade 2 on the Eastern Cooperative Oncology Group (ECOG) scale. Laboratory tests showed grade 1 secondary anemia, thrombocytosis (Table 1), and cancer antigen-125 (CA-125) levels of 520 IU/L.

Between June and October 2015, three cycles of Carboplatin (AUC 5) and Paclitaxel (175 mg/m^2^) were administered every 3 weeks, followed by two cycles of Carboplatin (AUC 5) alone due to poor clinical tolerance (asthenia) and hematological toxicity (thrombocytopenia grade 1 and anemia grade 2). A post-chemotherapy CT scan described a partial response according to the Response Evaluation Criteria in Solid Tumors (RECIST 1.1): tumor in the right ovary, 52/19/44 mm; tumor in the left ovary, 42/27/37 mm; no ascites; and one peritoneal nodule of 11/10 mm (Figure 2).

In December 2015, the patient underwent a total hysterectomy with bilateral adnexectomy, bilateral pelvic lymphadenectomy, and sub-umbilical parietal pelvic peritonectomy. The pathology report indicated high-grade ovarian serous carcinoma, ypT3aN0 G3. Hyperthermic intraperitoneal chemotherapy was also performed, but hospitalization was prolonged by over one month due to severe subsequent anemia (Hb 6.4 g/dL) requiring blood and plasma transfusions. The patient was discharged in a good clinical condition.

#### 2.1.6. Follow-Up

In April 2016, a CT scan (Figure 3) revealed progressive disease by a left external iliac lymph node of 30/29 mm, with discrete ureteral dilation and an inframesocolic peritoneal nodule of 15/44 mm. The tumor was considered platinum-sensitive, and between June and October 2016, the patient received three cycles of Carboplatin (AUC 5) plus Paclitaxel (175 mg/m^2^) and then one cycle of Carboplatin (AUC 5) alone due to side effects (allergic reaction to Paclitaxel, thrombocytopenia grade 1, and anemia grade 2); after cycle four, the patient requested interruption of chemotherapy.

In October 2016, a CT scan of the abdomen and pelvis showed a partial response according to RECIST v1.1 based on a decrease of 35% of the target lesions (the external iliac tumor mass now measured 26/18/31 mm and the inframesocolic nodule had disappeared). Laboratory tests showed a normal level of CA-125 and grade 3 anemia (Table 1).

Given the familial history and the fact that the patient had a partial response to platinum-based chemotherapy, genetic testing of the BRCA mutation was recommended. The genetic analysis report indicated a germline BRCA1 exon 2 mutation c.68_69delAG (p.GluValfs*17, del17q21.31), a pathogenic variant. The mutation was identified from peripheral blood with Next-Generation Sequencing and the multiplex ligation-dependent probe amplification technique [8].

Maintenance therapy with Olaparib 400 mg twice daily was started in November 2016 and continued throughout December 2023, with the best response of complete response on regular imaging assessments (Figure 4). The main side effects were asthenia grade 1 and nausea grade 1, successfully treated with metoclopramide 10 mg bid. At the last available clinical examination (December 2023), the patient had a good performance status (ECOG PS 0) and no symptoms. The visual summary of the outcomes over time is depicted in Figure 5.

The patient has no children or sisters, and her only brother was informed about his sister carrying the BRCA1 mutation after obtaining patient consent; genetic counseling was recommended for the patient’s family.

#### 2.1.7. Compliance

The patient had good compliance with the treatment regimens and subsequent follow-ups. However, due to iterative allergic reactions to Paclitaxel and hematological toxicity to Carboplatin, she could not tolerate doublet chemotherapy for the entire course prescribed in either of the two lines of treatment and even asked the medical team to interrupt the second line. Chemotherapy significantly affected the quality of life of this patient, also causing grade 3 asthenia and grade 3 fatigue.

### 2.2. Second Case Presentation

#### 2.2.1. Main Characteristics and Admission to Hospital

This report includes a case of a 55-year-old postmenopausal female presenting at the Surgical Oncology department of IOB (Institute of Oncology “Prof. Dr. Alexandru Trestioreanu”, Bucharest, Romania) with abdominal discomfort and vaginal bleeding in August 2015. She was diagnosed through a CT scan with a pelvic–abdominal tumoral mass following a routine gynecological consultation in February 2015, but she has refused further investigations (Figure 6).

#### 2.2.2. History, Comorbidities, and Interventions

The obstetrical history of the patient included four pregnancies resulting in two live births and two abortions. Her father had been diagnosed with lung cancer, with a rapid evolution leading to death, without oncological treatment and no further analysis of present mutations.

No other relevant conditions were reported.

#### 2.2.3. Examinations and Investigations

CT imaging of the thorax, abdomen, and pelvis was conducted in August 2015 and revealed a left ovarian tumor measuring 150/135/120 mm with multiple septations, cystic regions, and solid tumoral tissue, a normal-sized uterus displaced rightward and inseparable from the tumor, peritoneal carcinomatosis, a moderate amount of ascites, and a right pleural effusion occupying about one-third of the hemithorax, leading to atelectasis of the adjacent lung tissue due to compression (Figure 7).

#### 2.2.4. Diagnosis

Diagnostic laparoscopy was performed in August 2015. During the surgical intervention, a midline pubo-umbilical laparotomy uncovered a significant pelvic–abdominal tumor, accompanied by numerous tumor nodules scattered across both the parietal and visceral peritoneum. The patient also exhibited a moderate volume of hemorrhagic ascitic fluid. Multiple biopsies were conducted, and the initial histopathological findings indicated a malignant proliferation characteristic of poorly differentiated papillary adenocarcinoma, suggesting an ovarian or endometrial etiology. Immunohistochemical (IHC) testing corroborated the presence of high-grade serous papillary carcinoma with genital origins. It confirmed the ovarian source of the neoplasm, with positive estrogen receptors (60%), progesterone receptors (30%), a status of p53-positive (90%), and a Ki67 index of 40%. The pleural effusion was subjected to additional analysis, with positive cytological findings confirming the presence of neoplastic cells, leading to the classification of the case as FIGO stage IVA.

#### 2.2.5. Treatment and Outcomes

The Multidisciplinary Tumor Board, in accordance with NCCN guidelines, decided that the patient was a candidate for primary chemotherapy. She was admitted to the Medical Oncology department of IOB in September 2015, presenting with a good clinical and paraclinical status, ECOG 0. The CA-125 level was 1852 IU/L before chemotherapy.

Between September 2015 and February 2016, six cycles of Carboplatin (AUC 5) and Paclitaxel (175 mg/m^2^) were administered every 3 weeks, with good clinical tolerance and without hematological toxicity. A post-chemotherapy CT scan described a partial response according to RECIST 1.1. The CT imaging findings revealed a significant change in the patient’s tumoral burden, demonstrating a left adnexal tumor measuring 75/56 mm with malignant features and indistinct boundaries from the uterine fundus, suggesting direct invasion or close association. In a positive turn of events, the previously noted pleural effusion had resolved, and there was a complete disappearance of ascitic fluid, indicating a favorable response to the recent therapeutic intervention (Figure 8). After six cycles of chemotherapy, the CA-125 level was 9.04 IU/L.

In March 2016, the Multidisciplinary Tumor Board indicated interval debulking surgery, and the patient underwent a bilateral adnexectomy and omentectomy, which revealed several intraoperative findings. Notably, nodules indicative of peritoneal carcinomatosis, lumbo-aortic lymphadenopathy, and secondary liver lesions were detected (not observed on the preoperative CT scan). These findings were substantiated by histopathological examination, which confirmed carcinomatous infiltration of both ovaries and the omentum. After the surgery, the tumor marker CA-125 was measured at 7.04 IU/mL in May 2016.

#### 2.2.6. Follow-Up

The patient classified as FIGO IVA presented a partial response to the neoadjuvant chemotherapy regimen and was considered platinum-sensitive. She was proposed for another six cycles of chemotherapy with the same drug combination, Carboplatin (AUC 5) and Paclitaxel (175 mg/m^2^), starting in June 2016. During the patient’s chemotherapy regimen in September 2016, an adverse reaction to Carboplatin occurred during the fifth cycle of treatment. Manifestations included facial erythema and pruritus, while blood pressure remained within normal parameters, without respiratory symptoms or episodes of nausea or vomiting. The adverse effects were effectively managed with appropriate treatment, leading to remission of the symptoms. However, due to prolonged grade 1 thrombocytopenia, the medical team decided to discontinue the use of the chemotherapy protocol.

A significant result within the clinical narrative of this case, the detection of deleterious BRCA1 missense alterations in a compound heterozygous state, c.181T>G (p.Cys61Gly), in September 2016, set the stage for the introduction of Olaparib the following month. The therapeutic efficacy was confirmed via biannual CT scans of the thorax, abdomen, and pelvis, which indicated a positive outcome, achieving complete remission of the disease (Figure 9).

The patient’s treatment journey encountered a brief hiatus in October 2020 due to an asymptomatic SARS-CoV-2 infection, necessitating home isolation and a pause in Olaparib administration (due to pandemic restrictions). Treatment resumed in November 2020 after the patient tested negative for the virus. By December 2021, the patient maintained complete radiological remission, with CA-125 at normal levels. The values of the hemoglobin, thrombocytes, and CA-125 are displayed in the dynamics in Table 2.

The treatment with Olaparib continued until February 2022, when the MRI (magnetic resonance imaging) scans of the abdomen and pelvis divulged a pelvic recurrence of 4/3 cm with malignant features, adherent to the sigmoid colon, accompanied by a 2.5/2 cm nodule at the left uterine isthmus, presacral lymphadenopathies, and a right ureteral stone causing grade II hydronephrosis. The biological progression had been identified one month earlier based on a rise in the value of CA-125 (145.20 IU/L). In response, in March 2022, a significant surgical endeavor of secondary cytoreduction was performed, where the pelvic recurrence was excised, alongside a comprehensive hysterectomy, omentectomy, and pelvic and lumbo-aortic lymphadenectomy, supplemented by viscerolysis.

However, in April 2022, the patient developed a subocclusive syndrome, which was managed by subcutaneous drainage of the causative abdominal collection. The procedure was guided by CT imaging to position the drain tube in the left iliac fossa.

After complete recovery from the second cytoreductive surgery and following complications, the patient was initiated on treatment with Tamoxifen in November 2022, which has continued until the present, with good clinical status and without other recurrences of the disease. At the last available clinical examination (December 2023), the patient had a good performance status (ECOG PS 0) and no symptoms. The visual summary of the outcomes over time is depicted in Figure 10.

#### 2.2.7. Compliance

The adverse reaction to Carboplatin was the most notable treatment-related incident, which underscores the delicate balance between therapeutic efficacy and patient safety. It also emphasizes the need for vigilant monitoring of chemotherapy-related side effects.

Importantly, the patient’s tolerance to the PARP-inhibitor regimen was extremely good, presenting only low-grade hematological toxicities, thereby negating any need for treatment cessation or dosage adjustments. No adverse effects were reported during Tamoxifen treatment.

## 3. Discussion

### 3.1. Case Discussions

The first case is of a young nulliparous woman with BRCA1-mutated ovarian cancer with a strong family history of ovarian cancer and breast cancer. Our patient experienced significant side effects from chemotherapy; conversely, she had an excellent tolerance to Olaparib, which underlines the importance of tailored treatment approaches in patients with ovarian cancer and BRCA mutation. Although diagnosed at an advanced stage, the patient benefited from the long-term treatment with Olaparib, presenting a stable disease with a good quality of life.

The longevity of the second patient’s survival post-diagnosis, now extending to 100 months, marks a significant success in the treatment of ovarian cancer. Notably, a remarkable PFS of 65 months was achieved under Olaparib treatment. This PFS aligns with the growing body of evidence supporting the efficacy of PARP inhibitors in extending survival rates. The complete radiological remission maintained from August 2018 to February 2022 underlines the potential of the use of Olaparib as a maintenance therapy, which has redefined the paradigms of survivorship in ovarian cancer and represents a testament to the enduring potential of personalized oncology treatment offered by PARP inhibitors. This milestone prompted clinical deliberation regarding the duration of maintenance treatment with Olaparib, especially in the context of the patient maintaining complete remission, given the low risk of myelodysplastic syndrome and acute myeloid leukemia as a side effect of PARP inhibitors. Are patients with no evidence of disease still gaining benefit from Olaparib treatment in this setting? Further studies might investigate the optimal duration of the treatment due to well-known acquired resistance mechanisms.

### 3.2. Romanian Landscape of BRCA Mutations

Germline or somatic BRCA mutations are reported in 18–25% of patients with serous ovarian cancer [9]. It is known that patients carrying a germline BRCA mutation have a better prognosis [10]. Based on genetic registry data (ClinVar [10], OMIM [11]), the BRCA1 c.68_69delAG mutation (NM_007294.3(BRCA1):c.68_69delAG (p.Glu23Valfs)) identified in our first patient is a pathogenic variant (it induces frameshift substitution with subsequent loss of function) and is mainly associated with a founder effect. It was first described in the Ashkenazi Jews, and it is one of the most frequent mutations in this population (incidence of 4.16%, mostly in patients with breast cancer) [8,12,13,14]. Although the founder variant was observed in Ashkenazi Jews, there were also certain Arabic populations harboring this mutation. Moreover, is appears to be widely present in Asian populations (Azerbaijan, China, India, Iran, Iraq, Japan, Malaysia, Mongolia, Nepal, Pakistan, Philippines, Qatar, Russia, Saudi Arabia, Sri Lanka, Singapore, and Thailand) [15].

Currently, little is known about the full range of BRCA1/2 mutations in Romanian OC patients. Our first patient had a rare BRCA1 germinal mutation, possibly linked to the historical presence of Jewish populations in Romania, despite the patient not reporting Jewish ancestry. Alternatively, the c.68_69delAG mutation might have occurred independently due to genetic instability [16,17].

The first molecular investigation into the role of BRCA genes in breast and ovarian cancer within Romania was carried out in 2010, when 17 patients from unrelated hereditary breast and ovarian cancer families in northeastern Romania were screened for BRCA1 and BRCA2 mutations. Negură et al. identified four BRCA1 mutations and two BRCA2 mutations across these families, resulting in an overall mutation frequency of 41%. Although the number of screened patients was small, two mutations (BRCA1 c.2241dupC and BRCA2 c.8680C>T) were novel and not listed in the database [16]. Goidescu et al. reported the results of 130 cases of breast cancer, out of which 82 had either pathogenic/likely pathogenic mutations or VUS mutations. The most common BRCA1 variant identified was c.3607C>T (seven cases), followed by c.5266dupC and c.4035delA (each found in four cases). For BRCA2, the mutations c.9371A>T and c.8755-1G>A were identified in six cases, and VUS mutations were detected in three cases [17]. Studies in Eastern Europe have identified 185delAG and 5382insC as the most common BRCA1 mutations and 6174delT as the most frequent in BRCA2. The BRCA1 5382insC mutation is particularly prevalent in Central and Eastern Europe [18]. However, Negură et al. found no instances of this mutation in 170 breast and ovarian cancer patients with no family history [19,20]. In another study, Stănculeanu et al. reported that the most common BRCA1 mutation in high-grade serous ovarian carcinoma was c.5266dupC (5382insC), followed by c.3607C>T. The BRCA1 c.68_69delAG mutation was found in one patient (1.5%) [21]. Eniu et al. evaluated the presence of germline BRCA1/BRCA2 mutations in 250 high-risk breast cancer patients tested in one center. The BRCA1 c.68_69delAG was identified as a founder mutation in one case, but no other details were given about this patient [22]. Vidra et al. assessed 250 Romanian women with breast cancer and 240 with ovarian cancer using Next-Generation Sequencing, finding BRCA mutations in 47 breast cancer patients (63.83% BRCA1, 36.17% BRCA2) and 60 ovarian cancer patients (72% BRCA1, 28% BRCA2). The most frequent mutation was BRCA1 c.3607C>T, detected in 18 cases, followed by BRCA1 c.5266dupC (17 cases) and BRCA2 c.9371A>T (12 cases). The BRCA1 c.3607C>T mutation, though less common in Romania, was mainly linked to triple-negative breast cancer and ovarian serous adenocarcinoma [23]. In an ovarian cancer cohort, DNA sequencing identified sixteen deleterious mutations, including seven frameshift mutations, one single nucleotide variant, and four missense mutations. The most common BRCA1 mutations were c.3607C>T (26%), c.5266dupC (19%), and both c.1687C>T and c.181T>G (9% each). Notably, c.181T>G and c.1687C>T are predominant in the Slovenian population and common in southeastern Europe, while c.181T>G is also frequently found in Romania and Slavic countries [24].

A recent study on a cohort of 411 Romanian patients diagnosed with breast cancer showed that the c.3607C>T, c.181T>G missense variants, the c.5266dupC, c.68_69delAG (p.GluValfs) frameshift variants are recurrent in BRCA1 carriers. They identified eight cases of c.181T>G mutation and three of c.68_69delAG [25].

All studies on BRCA mutations in Romania have involved relatively small sample sizes, usually from a few dozen to a few hundred patients. The lack of a national registry for hereditary breast and ovarian cancer limits comprehensive data collection, making it difficult to provide accurate national statistics. Consequently, we rely on extrapolating from the few available studies. This highlights the urgent need for a national registry to better track BRCA mutations and enhance our understanding of their prevalence and impact across the country.

### 3.3. Biomarkers in Ovarian Cancer–BRCA and ER

Based on the first patient’s characteristics and data from the landmark Study 19, which showed a significant median PFS of 8.4 months for Olaparib vs. 4.8 months for placebo, *p* < 0.0001 [8,26], the medical team opted for the treatment with Olaparib. In Study 19, a small subset of fifteen patients received Olaparib for more than 6 years; three of them had BRCA1 mutations [27]. Our patient obtained a seemingly similar benefit and outcome from Olaparib (PFS of 69 months in December 2023 for the first patient), highlighting BRCA1 (and BRCA2) mutations as a biomarker for a particularly long benefit from Olaparib, albeit in a limited population (one-third of cases in Study 19); we speculate that some individual mutations might be of particular importance in this respect. No data are available about the potential role of BRCA1 c.68_69delAG or other genetic variants as a favorable prognostic factor, but this hypothesis is worth considering in further, more expanded genetic mapping research. The PFS of 65 months for the second patient surpassed the median overall survival (mOS) of 51.7 months reported in the pivotal SOLO-2 trial, in which Olaparib was administered to patients with relapsed high-grade serous or endometrioid ovarian cancer who were platinum-sensitive [28].

Lheureux et al. analyzed the clinical and molecular data of the long-term responders (response durations of more than 2 years to Olaparib) versus short-term responders (response of less than 3 months to Olaparib) [29]. Extensive molecular profiling demonstrated that durable long-term responses were multifactorial and driven by germline and somatic BRCA1/2 mutations. The biomarker indicating which patients would benefit the most from Olaparib was homologous recombination repair deficiency (HRD), with enrichment for mutations in BRCA2. Although most studies indicate that BRCA2 mutations are associated with prolonged survival in invasive epithelial ovarian cancer [30], we presented a case of a patient with BRCA1 mutation and extended survival benefit. Our patients were not tested for HRD, as it was not reimbursed in Romania at that moment. The literature lacks significant data regarding the clinical importance of the coexistence of BRCA mutations and homologous recombination deficiency in ovarian cancer. From a biological perspective, their concurrence could potentially explain the increased efficacy of PARP inhibitors in this patient group.

In our endeavor to seek answers, we analyzed studies investigating the efficacy of other PARP inhibitors, like rucaparib. A pooled analysis across ARIEL2 and Study 10 provided a characterization of patients with high-grade recurrent ovarian carcinoma who had a long-term response to rucaparib [31]. A long-term response, defined as lasting over a year, was achieved in 28% of responders, with 11.6% experiencing a response longer than two years. Deleterious BRCA mutations, particularly small insertions or deletions, were common across all subgroups, but homozygous deletions were more frequent in long-term responders. This suggests that certain BRCA structural variants, like homozygous deletions, may lead to better outcomes. Secondary somatic reversion mutations can restore BRCA function and contribute to PARPi resistance, but structural variants—accounting for 10–15% of germline BRCA mutations—are less prone to these reversions, possibly explaining the durable responses observed (up to four years). It remains to be studied whether this applies to patients with BRCA mutations treated with Olaparib. Additionally, BRCA Ashkenazi Jewish founder mutations (BRCA1 c.68_69delAG, BRCA1 c.5266_5267insC, or BRCA2 c.5946delT) were detected at similar rates across all subgroups, highlighting the need for further research to understand their role in tumor response to PARP inhibitors [30,31].

In a retrospective cohort study, Zhang et al. linked three clinical factors to prolonged PFS in patients with platinum-sensitive recurrent ovarian cancer under treatment with Olaparib: CR to the last platinum-based therapy, PFI longer than 12 months, and BRCA mutant type [32]. This hypothesis is strengthened by the results of a 5-year follow-up of the SOLO1 study, which associated CR at baseline with a decrease in the risk of recurrence or death by 63% [33]. Our first patient had a platinum-free interval of 8 months and just a partial response to the last sequence of chemotherapy. Hence, we can assume she presented other features than the BRCA1 mutant type that led to the prolonged response, but this has to be explored to a greater degree.

With regards to the future, OLALA (NCT02489058), an ongoing observational study of long-term responders (at least 1.5 years) to Olaparib, collected samples from patients involved in several clinical trials using Olaparib as an investigational drug [34]. Outcomes will most certainly identify additional molecular characteristics capable of predicting prolonged susceptibility to Olaparib.

Despite the initial efficacy of PARP inhibitors (PARPi) in treating BRCA-mutated ovarian cancers, the development of resistance remains a significant clinical challenge. Several mechanisms have been implicated in mediating this resistance. One primary pathway involves secondary or “reversion” mutations in BRCA1 or BRCA2 genes that restore the functional reading frame, thereby reinstating homologous recombination repair capabilities in cancer cells. Additionally, alterations in drug efflux transporters, such as upregulation of P-glycoprotein, can reduce intracellular concentrations of PARPi, diminishing their effectiveness. Another resistance mechanism includes the loss of 53BP1 or other shieldin complex components, which alters the balance between homologous recombination and non-homologous end-joining, favoring DNA repair pathways that circumvent PARP dependency. Furthermore, increased stabilization of replication forks through upregulated ATR/CHK1 signaling or other factors can also contribute to resistance by protecting stalled replication forks from degradation. Understanding these diverse resistance mechanisms is crucial for developing combination therapies or novel agents that can overcome or circumvent PARPi resistance [35].

A meta-analysis of 17 studies on the impact of estrogen receptor (ER) expression on the prognosis of ovarian cancer showed that higher expression of ERα and ERβ is linked to better survival in ovarian cancer. Still, results vary significantly based on the choice of ER antibody clones used in IHC analysis [36]. Since its introduction into clinical practice in the early 1980s, the selective estrogen receptor modulator Tamoxifen has been used to treat ovarian cancer. Overall mean response rates to this therapy have been reported to be 10–15%, with a disease stability rate of 30–40% [37]. In these clinical trials, most patients had received extensive pretreatment, and several of the studies did not select for ER-positive cases. The use of Tamoxifen in trials where at least 50% of the patients had not received more than one prior treatment showed an overall response rate of 26%, with a 9% complete response rate, as noted by Perez-Gracia et al. [38]. These results were in contrast to clinical studies of heavily treated patients, whose response rate was only 4%. Letrozole and Tamoxifen had similar overall responses (8% and 11%, respectively) and clinical benefit rates (41% and 33%, respectively) in a 25-year study conducted on 269 HGSOC patients [39]. The most probable patients to benefit were those with a prolonged treatment-free time and a high ER score, like our second patient. The potential implications of ER-positive and ER-negative statuses in the context of BRCA-mutated ovarian cancer and how these interactions might impact treatment outcomes should be further investigated, as the available data are not as strong, as in the case of breast cancer (where ER-positive BRCA-mutated breast cancer is a special subgroup of patients with poor prognosis).

### 3.4. Limitations of the Study

We acknowledge that the conclusions drawn from these case reports are inherently limited due to the small sample size (two) and the lack of broader generalizability. Case reports, by nature, provide detailed insights into individual patient outcomes but do not offer the statistical power needed to establish definitive cause-and-effect relationships or to predict outcomes in a larger population. Therefore, while these cases provide valuable information, further studies with larger cohorts are necessary to validate our observations and draw more robust conclusions.

## 4. Conclusions

This first case illustrates that patients with advanced serous high-grade ovarian cancer with a strong family history and BRCA1 mutation can have a long-term derived benefit under Olaparib treatment, with good quality of life and prolonged progression-free survival and overall survival, even in the context of poorly tolerated chemotherapy.

The second case exemplifies the complexities and potential triumphs of managing a case of advanced ovarian cancer with BRCA1 mutation, with the transformative impact of PARP inhibitors in oncology. The individualized treatment approach, driven by genetic profiling and responsive to the dynamic changes in disease status, has enabled an extension of survival and the maintenance of quality of life. The case further highlights the necessity for an ongoing multidisciplinary approach and the adaptability of the treatment plans in response to patients’ evolution, recurrences, and complications. The role of surgery (interval debulking surgery and secondary cytoreduction) in ovarian cancer remains a cornerstone in the management of the disease, providing a clear survival benefit when optimally executed and integrated with systemic therapies, such as chemotherapy, targeted treatments like PARP inhibitors, and hormonal therapy, such as Tamoxifen.

Data from the literature are scarce regarding predictive factors for long-term responders. In this paper, we propose some explanations regarding the subcategory of patients who received Olaparib.

## Figures and Tables

**Figure 1 diagnostics-14-01898-f001:**
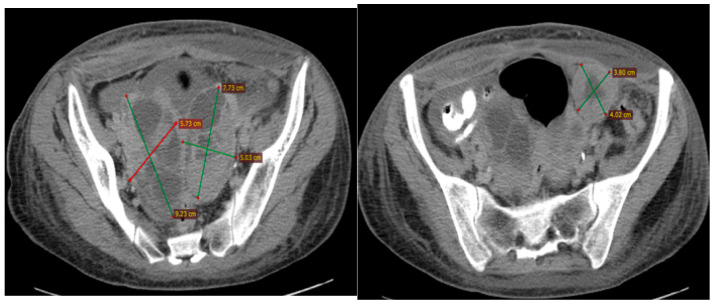
The baseline CT scan of the pelvis from June 2016 showing the ovarian tumors and the lymph node in the left iliac fossa.

**Figure 2 diagnostics-14-01898-f002:**
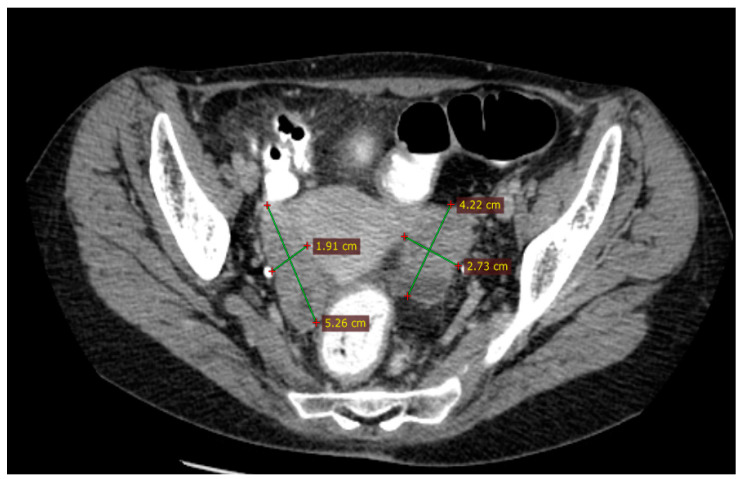
The first follow-up (CT scan of the pelvis, October 2015) showing partial response to treatment.

**Figure 3 diagnostics-14-01898-f003:**
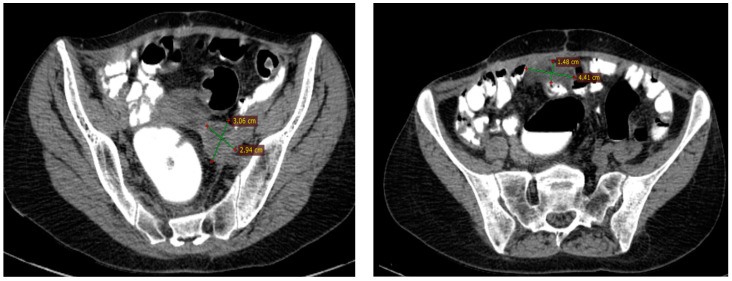
Second radiological assessment: April 2016 CT scan showing progressive disease with a new lesion—inframesocolic nodule of 15/44 mm.

**Figure 4 diagnostics-14-01898-f004:**
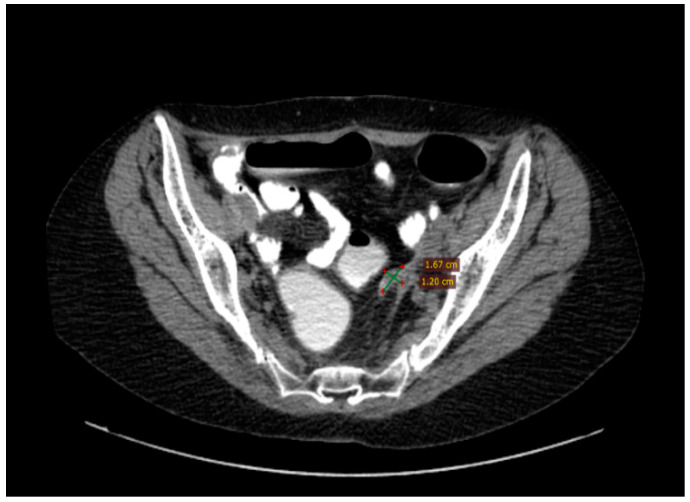
The complete response was assessed based on a CT scan from February 2020.

**Figure 5 diagnostics-14-01898-f005:**
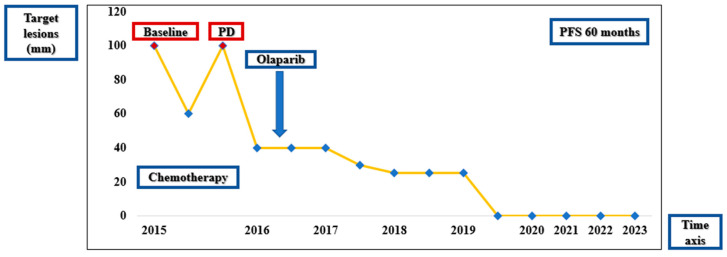
Timeline showing the pattern of response according to RECIST 1.1 and CA-125.

**Figure 6 diagnostics-14-01898-f006:**
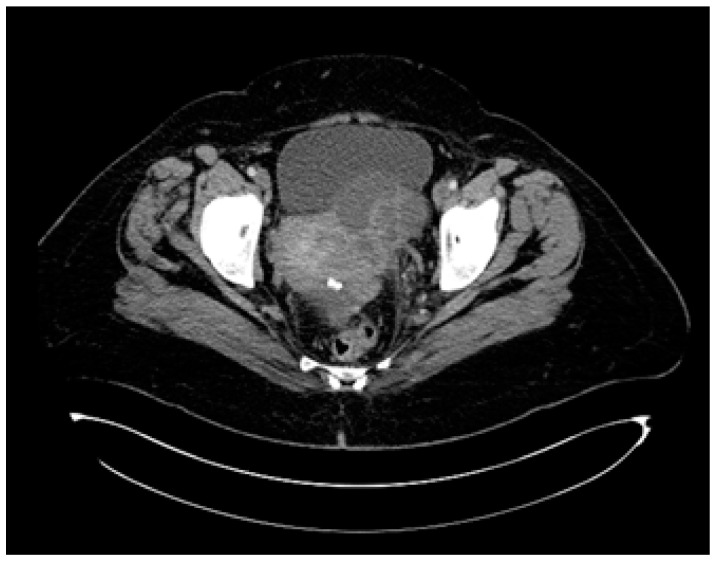
Pelvic CT scan from February 2015, six months before presentation at the oncology department.

**Figure 7 diagnostics-14-01898-f007:**
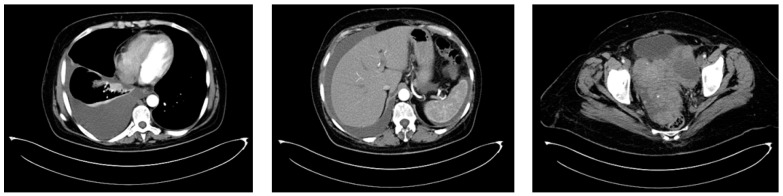
The baseline CT scan of the thorax, abdomen, and pelvis from August 2015 showing the left ovarian tumor, peritoneal carcinomatosis, ascites, and right pleural effusion, leading to atelectasis of the adjacent lung tissue.

**Figure 8 diagnostics-14-01898-f008:**
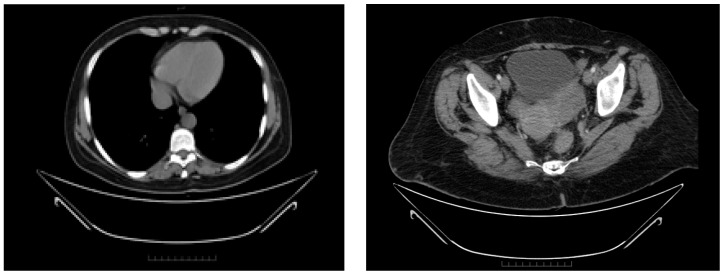
The first follow-up (CT scan of the thorax and pelvis, March 2016) showing partial response to treatment.

**Figure 9 diagnostics-14-01898-f009:**
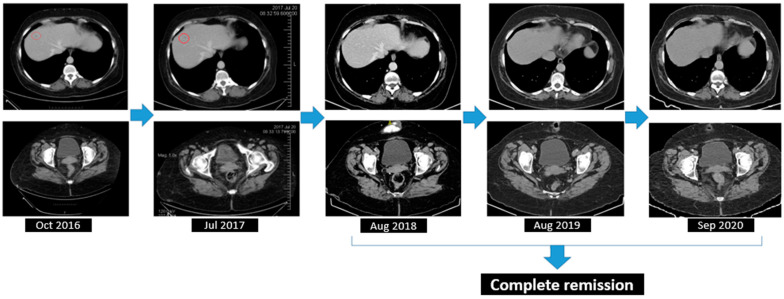
Complete response assessed through CT from August 2018 to September 2020.

**Figure 10 diagnostics-14-01898-f010:**
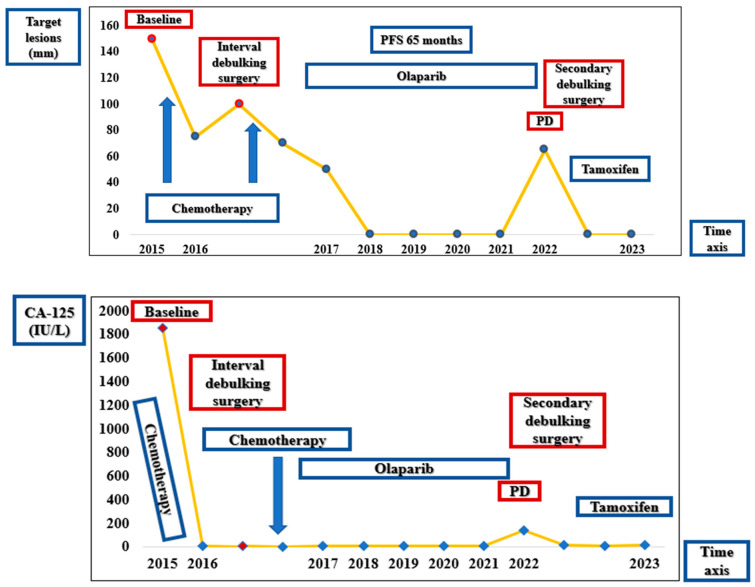
Timeline showing the pattern of response according to RECIST 1.1 and CA-125.

**Table 1 diagnostics-14-01898-t001:** Evolution of hemoglobin, thrombocyte count, and CA-125 levels. (Lowest values during the corresponding time interval).

Time Point	Hemoglobin (g/dL)	Thrombocytes (mm^3^)	CA-125 (IU/L)
July 2015 (diagnosis)	10.4	883,000	520.00
December 2015 (3 × Carboplatin/Paclitaxel; 1 × Carboplatin)	6.4	238,000	10.00
April 2016 (progression)	11.9	190,000	354.00
November 2017 (Olaparib start)	10.6	125,000	12.00
April 2018–January 2020 (stable disease under Olaparib therapy)	13.1	317,000	8.00
February 2020–February 2021 (complete response under Olaparib therapy)	13.8	293,000	7.80
March 2021–March 2022	12.4	212,000	7.20
March 2022–December 2023	13.4	233,000	7.60

**Table 2 diagnostics-14-01898-t002:** Evolution of hemoglobin, thrombocyte count, and CA-125 levels. (* Lowest values during the corresponding time interval).

Time Point	Hemoglobin (g/dL)	Thrombocytes (mm^3^)	CA-125 (IU/L)
September 2015 (diagnosis)	10.0	552,000	1852.00
February 2016 (after 6 × Carboplatin/Paclitaxel)	11.5	296,000	9.04
May 2016 (after interval debulking surgery)	11.5	233,000	7.04
September 2016 (after 5 × Carboplatin/Paclitaxel)	8.0	96,000	-
October 2016 (start Olaparib maintenance)	7.9	121,000	5.79
October 2016–August 2018 (partial response under Olaparib therapy)	9.5 *	256,000 *	7.80 *
August 2018–December 2021 (complete response under Olaparib therapy)	9.7 *	330,000 *	12.70 *
January 2022 (progression)	11.3	278,000	145.20
June 2022 (after secondary cytoreductive surgery)	10.1	488,000	18.65
November 2022 (start Tamoxifen treatment)	11.7	229,000	8.66
November 2022–December 2023	12.7 *	190,000 *	16.37 *

## Data Availability

The raw data supporting the conclusions of this article will be made available by the authors on request.

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
