# Peer review of "Long-Term Survival in BRCA1 Mutant Advanced Ovarian Cancer: Unveiling the Impact of Olaparib"

_diagnostics, 2024, doi:10.3390/diagnostics14171898_

Round 1

Reviewer 1 Report

Comments and Suggestions for Authors

The study reports the long-term survival of 2 ovarian cancer patients with BRCA1 pathogenic mutations by Olaparib treatment. Decades’ study from both fundamental science and clinical trials have demonstrated that PARP inhibitor are the powerful drug to treat BRCA mutated ovarian cancer. The report provides two ovarian cancer cases from Romania treated by Olaparib. The outcome is expected to be effective.

Comments:

1.     Patient one is a carrier for BRCA1 c68_69delAG. Authors described this variant as “rare germinal mutation”. This is not correct. BRCA1 c68_69delAG is not only as founder mutation and highly prevalent in Ashkenazi Jewish populations, but also widely present in different ethnic populations, like Indian, Pakistani, and Arabic population  (PMID: 36385461);

2.     The manuscript describes “Currently, little information is available on the full spectrum of BRCA1/2 mutations in Romanian OC patients; moreover, our first patient harbored a particularly rare germinal mutation of BRCA1”. However, multiple BRCA studies have been published in recent years (PMID: 35409996, PMID: 37239058, PMID: 38674216, PMID: 20567915, PMID: 29785153 etc). Further, in the Discussion part, to much discussion for BRCA mutation and PARP and ovarian cancer, of which the relationship has been well determined by extensive studies and not directly related with the focus of the study. I suggest to significantly replace such discussion with the BRCA mutation and cancer in Romania patients and published BRCA data.

Comments on the Quality of English Language

ok

Reviewer 2 Report

Comments and Suggestions for Authors

This case report provides valuable insights into exceptional long-term responses to Olaparib in BRCA1-mutated advanced ovarian cancer. The detailed presentation of two cases with different BRCA1 mutations achieving remarkable progression-free survival of 69 and 65 months, respectively, is noteworthy. These outcomes can contribute meaningfully to general understanding of PARP inhibitor efficacy. However, there are several areas that require improvement to enhance the scientific quality and clarity of the report.

 Major comments:

INTRODUCTION:

 Page 1 L40 - L50:

The introduction lacks a clear statement of the objective or purpose of this case report. The authors should explicitly state why these cases are unique or significant enough to warrant publication.

L54 and L142: Instead of stating “We report a case….” The authors can use the phrase “This report includes a case ………”

L152: Father of case 2 was diagnosed with Lung cancer. Including any information about the mutations and length of survival of the father would be relevant. 

L259-260: Rephrase as “....with strong family history of ovarian cancer and breast cancer…”

DISCUSSION:

In the discussion authors MUST limit the number of smaller paragraphs and combine them as contextually relevant larger paragraphs. Smaller paragraphs give the impression the report is poorly edited manuscript.

Lengthy review of mutations and its prevalence is irrelevant to this case report. It must be shortened and can include mutations or aberrations found in the patients included in this case report. Also, historical account of Ashkenazi Jewish migration and settlement need to be narrowed to the region of the patients or to any connection of the patients to the settlers in the family tree may be more appropriate.

The discussion section attempts to contextualize the presented cases within the broader landscape of BRCA-mutated ovarian cancer treatment. However, the flow of ideas is somewhat disjointed, jumping between topics without clear transitions. A more structured approach, perhaps organizing the discussion around key themes (e.g., genetic factors, treatment responses, implications for clinical practice) would enhance the coherence and will make it easy to read and understand.

While the discussion does reference the specific cases presented, it often fails to fully integrate these findings into the broader scientific context For example 

Page 13 L 382 - L390:  

The authors mention the Lheureux et al. study, which analyzed the long-term responders to Olaparib, but they did not directly compare the characteristics or outcomes of the cases to the findings from that study. The authors mention homologous recombination deficiency (hHRD) with enrichment mutation in BRCA2 as a biomarker for predicting Olaparib response. Though the patients discussed in this case report has BRCA1 mutation, the authors don't discuss whether their cases were tested for hHRD or how this might relate to their long-term response.

 Page 13 L422 - L430: 

BRCA structural variants: While the authors discuss the specific BRCA mutations found in their cases, they don't link their findings to the context of durability and potential resistance mechanisms.

Page 14 L446 – L461:

The discussion touches on the impact of estrogen receptor (ER) status in ovarian cancer prognosis, but doesn't fully explore its relevance to PARP inhibitor efficacy. They authors can briefly discuss about how the ER status might interact with BRCA mutation status in determining response to Olaparib?

Limitations: The discussion lacks a clear acknowledgment of the limitations inherent in drawing conclusions from case reports.

L464: Rephrase the sentence. Indicating as a Strong Family History would be appropriate. 

L479 - 481: Revise the last paragraph. For example, it can be broken into two sentences and instead of stating “..........that is why we proposed…..” it can be “However, this paper outlines …..”. This would require the removal of However, from the beginning of the paragraph. 

Minor comments: 

There are occasional grammatical and typographical errors throughout the manuscript that should be corrected

L305: What do authors mean by ‘Moldavian fairs’ do they mean? Are they alluding to some temporary events leading to long term inheritance of mutations?

L324: Leave a space after BRACA1

Page 12 L 333: typo error “Demographic and clinical characteristICs in paTients with ovarian..”

Page 13 L398: spelling error “waere”

OVERALL:

This case report offers important observations on the possibility of prolonged survival in certain BRCA1-mutated ovarian cancer patients treated with Olaparib. It emphasizes the necessity for additional studies to identify predictive markers for sustained PARP inhibitor efficacy and reinforces the value of tailored treatment strategies in cancer care. However, the manuscript fails to discuss about the status of the biomarkers that indicate resistance. For example, BRCA1/2-associated tumors are reported to be highly sensitive to platinum compounds and poly (ADP-ribose) polymerase inhibitors; However, they eventually acquire resistance to this type of therapy. Therefore, any available information related to PARP would add more value to this case report. 

It is highly recommended to incorporate the revisions suggested above. Incorporating the suggested improvements to the report would offer a more nuanced understanding of these cases and the therapeutic outcomes.

Comments on the Quality of English Language

In the discussion authors MUST limit the number of smaller paragraphs and combine them as contextually relevant larger paragraphs. Smaller paragraphs give the impression the report is poorly edited manuscript.

Reviewer 3 Report

Comments and Suggestions for Authors

The manuscript by Afrasanie is a report of two cases of ovarian cancer being treated by olaparib. Although the cases are interesting, there have been many studies performed to date supporting the use of Olaparib in women with BRCA mutations in ovarian cancer. Hence, there is little new information provided by the current reports.

Round 2

Reviewer 1 Report

Comments and Suggestions for Authors

The revision substantially improved the overall quality.

Reviewer 3 Report

Comments and Suggestions for Authors

I have already reviewed the original version of the paper and there is nothing new in the revised manuscript that would lead me to change my opinion.